# MP-GCN: A Phishing Nodes Detection Approach via Graph Convolution Network for Ethereum

Tong Yu [ID], Xiaming Chen, Zhuo Xu [ID] and Jianlong Xu *[ID]

College of Engineering, Shantou University, Shantou 515063, China; 20tyu@stu.edu.cn (T.Y.); chenxm@stu.edu.cn (X.C.); 20zxu3@stu.edu.cn (Z.X.)
* Correspondence: xujianlong@stu.edu.cn

**Abstract:** Blockchain is making a big impact in various applications, but it is also attracting a variety of cybercrimes. In blockchain, phishing transfers the victim's virtual currency to make huge profits through fraud, which poses a threat to the blockchain ecosystem. To avoid greater losses, Ethereum, one of the blockchain platforms, can provide information to detect phishing fraud. In this study, to effectively detect phishing nodes, we propose a phishing node detection approach as message passing based graph convolution network. We first form a transaction network through the transaction records of Ethereum and then extract the information of nodes effectively via message passing. Finally, we use a graph convolution network to classify the normal and phishing nodes. Experiments show that our method is effective and superior to other existing methods.

**Keywords:** graph convolution network; Ethereum; phishing fraud; message passing

## 1. Introduction

Blockchain is a distributed, open, public ledger that records information pertaining to transactions between two parties. It has garnered attention owing to its anonymity, non-repudiation, non-tamperability, and permanent storage [1,2]. In 2009, Satoshi Nakamoto launched the Bitcoin project, the first large-scale application of blockchain, and was the first to introduce cryptocurrency to the real world [3,4].

Ethereum is one of the world's largest smart contract platforms, and Ether is the second-largest cryptocurrency in the world [3]. The anonymity of Ethereum allows cybercrime to occur easily because they cannot be convicted for cybercrimes in the real world easily [3]. Most notably, the Decentralized Autonomous Organization (DAO) hack in 2016 and the Parity Wallet hack in 2017 together caused more than $400 million in damages [5]. Phishing, a type of anomaly scoring network, is a cyber-fraud where a trusted organization (online bank, online retailer and so on) is impersonated to send fake emails and provide fake websites to users [6]. The victims often reveal their private information, such as credit card numbers, bank accounts, and ID numbers [7]. Typically, a phishing attack begins with an email sent to a victim that appears to be from a real organization [8]. Furthermore, this type of fraud can also happen in Ethereum through the transfer of irrevocable Ether through victims to the fraudulent sides [9]. In 2017, the Chainalysis Team were reported to account for more than 50% of all cyber-crimes in Ethereum [10]. The most representative example is the chess organization "Bee Token" incident in 2018, in which scammers defrauded approximately $1 million in 25 h [11]. To create a favorable investment environment for the Ethereum ecosystem, the Ethereum community must urgently identify an effective method for detecting and preventing phishing fraud.

Previously, traditional phishing scammers disguised themselves as credible entities to steal users' sensitive information, such as usernames, passwords, and bank card information, such that they can then steal users' property [7]. Consequently, several methods of phishing detection have been proposed to reduce property loss, such as detecting the con-

tent of the emails and the information on the websites [12]. It is noteworthy that phishing fraud on Ethereum exhibits certain characteristics different from traditional phishing.

First, phishers must exchange the illegally obtained cryptocurrency with real currency to gain profit. Second, the transaction records in Ethereum are publicly accessible, which provides a complete data source for determining the transaction methods of different Ethereum users, thereby facilitating phishing node detection. Finally, most traditional phishing frauds rely on emails and websites to obtain sensitive user information, whereas phishing frauds on Ethereum rely on more diverse sources, which allow phishing information to be spread in many forms [13].

Therefore, traditional phishing detection methods are insufficient to solve the phishing problem in Ethereum.

We can see phishing network transactions in Ethereum occur in a certain direction, as shown in Figure 1. The phishing node is typically a node that accepts input from multiple nodes in a short duration and then transfers the amount to a specific node or nodes after obtaining a significant amount of money to transfer the stolen money rapidly and withdraw it, whereas the normal node interacts more frequently with its neighboring nodes and trades the received money with its neighbors. No reverse interaction with input nodes is becoming a visual feature of phishing nodes. Figure 1 is the transaction flow plotted from the first-order neighborhood nodes, which are directly connected nodes with an edge. We drew it based on our experimental data.

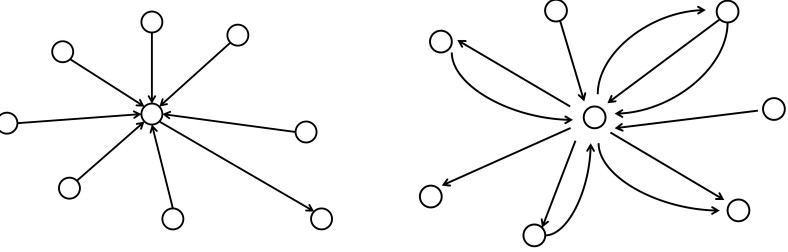

Transaction flow of phishing network　　　　　　Transaction flow of normal network

**Figure 1.** Direction of phishing and normal network transactions.

In the ethereum network, a node represents an address. In practice, if a user wants to transact with a node, either the user queries whether it is a phishing node from a list published by a third-party authority, or the user is completely unaware of it. In the case of not knowing whether the node is a phishing node, we determine whether the node is a phishing node by inputting the transaction network consisting of the destination node and his first-order or second-order neighbors. The experiment improves the timeliness of detecting phishing nodes. In addition, there is no need for users to wait for the authority to release the latest list of phishing nodes.

Owing to the open and transparent nature of Ethereum, information from transaction records can be extracted [4,14], thus making it possible to detect phishing fraud on the Ethereum network. In the Ethereum network, each node has a unique address, and linking between nodes with edges implies the existence of at least one transaction between two nodes. However, the use of transaction records from the Ethereum network for phishing detection is restricted by the following aspects:

- Imbalanced data: According to etherscan.io, a well-known block platform for Ethereum to help exploration and analysis by listing all for all transactions [15]. The total number of addresses and transactions in Ethereum exceeds 500 million and 3.8 billion, respectively, whereas the total number of labeled phishing addresses posted on etherscan.io is only 2041, with disparity between positive and negative nodes comparisons [16].
- Heterogeneous Network: Some transactions are associated with public or popular addresses, such as wallets, exchanges, and famous Initial Coin Offering (ICO) addresses which aim to raise funds to solve other problems [11], whereas many are

personal addresses with less than five transactions and transaction amounts. In such a heterogeneous network, phishing nodes are difficult to distinguish from normal nodes using only network topology information.

- Extract effective node representation: Blockchain phishing detection can be considered as a binary classification problem in machine learning, where the classification performance is related to the representation validity of the nodes. In addition, we have 1262 phishing nodes as positive labels and 1222 normal nodes as negative labels; thus, we can use supervised learning to fit the data and the labels during training the model.

Owing to the issues described above, we propose the message passing based graph convolutional network (MP-GCN) to detect phishing fraud in Ethereum networks. First, we build a phishing network with labels by using the transaction records. Second, we randomly initialized the node embedding data. To obtain graph feature, we used graph embedding [17], which is a method for generating the embedding representation containing structural information through low dimensions and is the basic component that connects the features with digital representation [18]. Graph embedding represents nodes in a continuous vector space [19]; thus, we can obtain the node embedding of every node. We use node embedding to represent the features of the nodes for nodes classification as phishing detection. After obtaining the embedding of each node, we use Graph Convolutional Network (GCN) as the supervised learning method to train our model. We fit the labels to the node embedding so that each node embedding corresponds to the label to obtain a good model classification. Finally, phishing node detection is performed using the trained model to classify test nodes into phishing nodes and normal nodes.

The statement of purpose is as follows:

(1) We propose a method aiming at phishing scam detection on Ethereum using MP-GCN, and this method outperforms existing methods in phishing detection problem.

(2) We design a new graph embedding model for the transaction network that takes into account the topological information of the network and the features of the nodes.

(3) We discuss the impact of different model parameters, model structure in the classification results in fishing detection problem.

(4) Our experiments provide a way to perform phishing detection on unknown nodes, and users can determine faster whether the node is normal or not without waiting for a third-party organization to publish a list of phishing nodes.

The remainder of this paper is organized as follows: In Section 2, we present recent studies regarding phishing fraud and malicious activities detection in Ethereum and anomaly detection based on graph embedding. In Section 3, we define the method to identify phishing fraud on Ethereum and present a comprehensive description of MP-GCN. In Section 4, we describe the data used in our experiments and evaluate the performance of our model on phishing node detection. In Section 5, we discuss our experiments and future endeavors. In addition, in Section 6, we conclude this paper.

## 2. Related Work

One kind of phishing detection methods is based on the contents of websites and email messages, where machine learning is used to extract the features from such sources for phishing detection. Zouina and Outtaj [20] proposed a phishing website detection system using a support vector machine (SVM) with a multilabel classifier. Priya et al. [21] proposed a plain Bayesian classifier for phishing website detection. Various types of features were used, such as source codes, URL features, and image features. Sahingoz [22] proposed a random forest algorithm based on Natural Language Processing (NLP) features for the phishing detection of URLs. Zouina et al. [20] used only six URL features and their similarity index as input features for phishing website detection systems. Moghimi [23] used the features of URLs and page elements for website evaluation. Traditional phishing detection relies significantly on the content, where detection becomes complicated if the

inaccessible website content is encountered or the propagated information on the emails is not available.

Another method then relied on non-textual information. Abdelnabi et al. [24] compared the similarity of web screenshots to detect a phishing website; this method requires a large storage space to store screenshots of websites, and when multiple websites with the same URL appear, the first one that appears is considered legitimate. Wu et al. [25] used the Trans2vec scheme for phishing detection using transaction features in the Ethereum transaction network, constructed a transaction network based on Ethereum transaction data, obtained transaction features based on graph embedding techniques, and finally classified nodes using SVMs. Wu et al.'s scheme only utilizes the features of the neighboring nodes considered; it does not account for the network overall topological information and cannot fully utilize the node features in the transaction network. Wu et al. [26] further proposed T-EDGE with the main improvement of ensuring the same sequence of nodes during random wandering and solving the problem of multiple edges in the network. Shucheng et al. [27] proposed a Self-supervised Incremental deep Graph learning model (SIEGE) for phishing scam detection problem on Ethereum. In Shucheng et al.'s model, two pretext tasks designed from spatial and temporal perspectives help to learn useful node embedding from the huge amount of unlabelled transaction data.

At this moment, some blockchain anomaly detection methods have been proposed. Vatsal et al. [28] used a One-Class Graph Neural Network-based anomaly detection framework for detecting anomalies in the Ethereum blockchain network. Sayadi et al. [29] used One-Class Support Vector Machine (OCSVM) used planar separation of normal and abnormal data in a high-dimensional feature space, which is eventually used to identify anomalies in the Bitcoin network. Chen et al. [30] used graphical analysis to describe three main activities in Ethereum, including transfers, smart contract creation, and smart contract invocation, which are ultimately used by graph-based algorithms to detect accounts that exhibit anomalous behavior. Teng et al. [31] used a data slicing algorithm to distinguish between different types of contracts, which can be applied to anomaly detection and malicious contract identification. Aziz et al. [32] proposed the Light Gradient Boosting Machine (LGBM) method for accurate detection of fraudulent transactions for fraud detection in ethereum. Yun et al. [33] proposed an Early-stage Phishing Detection Framework containing data processing, feature extraction and detection components for early-stage phishing scams.

Recently, several effective anomaly detection methods based on graph embedding have been proposed. Liu et al. [34] proposed log2ve by converting log entries into a heterogeneous graph, representing each log entry into a low-dimension vector, separating malicious and benign log entries into different clusters, and identifying malicious ones. Li et al. [35] proposed AddGraph to detect anomalous edges in a dynamic graph by using a method based on extended temporal GCN with an attention model. Cai et al. [36] proposed StrGNN by using GCN and Sortpooling layer to extract the feature and use Gated Recurrent Units (GRU) to capture the temporal information for detecting anomalous edges in dynamic graphs.

## 3. Method

In this section, we first define the problem in Ethereum phishing detection and then introduce the proposed phishing detection framework. Subsequently, we introduce message passing for extracting features of nodes as the nodes embedding and GCN for phishing node classification. Finally, we introduce our algorithmic process.

### 3.1. Problem Definition

The transaction network can be represented as $G = (V, E, Y)$, where $G$ represents network graph, $V$ is a set of nodes, $E$ is a set of edges, and $Y$ is a set of labels. In the Ethereum transaction network, each edge contains the transaction amounts and timestamps. A larger transaction amount means a bigger edge weight if the transaction amount is used as the

edge weight. Similarly, a bigger timestamp proves that the transaction occurred later and the weight of the edge will be bigger if the timestamp is used as the edge weight.

The goal of phishing detection is to detect phishing accounts accurately; this can be modeled as a binary classification problem in which +1 represents phishing and $-1$ represents normal accounts. After the embedded values are obtained for each node, they are input into the model, and the labelled nodes are classified using the binary classification method, while the effects of non-labelled nodes on the experimental results are disregarded.

### 3.2. Phishing Detection Framework

To solve the phishing detection problem, we propose a framework, depicted as the flowchart in Figure 2. The phishing detection framework proposed herein comprises the following main components:

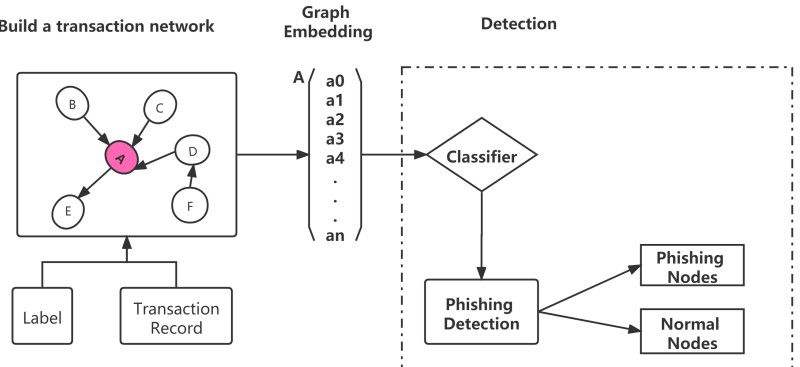

**Figure 2.** Framework of Phishing detection.

(1) Data Acquisition: Based on transaction records, we set the addresses are labeled as phishing addresses and normal addresses. These nodes are connected to their first-order neighbor nodes by edges. The first-order neighbor nodes of a vertex are the nodes directly connected to it with an edge, which in this experiment is a node that makes at least one transaction with the central node. A transaction record can be described as a quadruplet (S, R, T, B), which indicates the transaction initiator, i.e., a sender (S), who sends ethers (B) to the receiver (R), at time (T).

(2) Build the transaction network: Based on the data obtained, we build the transaction network by the transaction records because neighboring nodes will have similar embedding [37]. Every labeled node is connecting its first-order neighbor nodes with an edge. Each node is labeled, and each edge contains the transaction amount.

(3) Graph embedding: Graph embedding is an approach to transform nodes, edges, and their features into vector space (a lower dimension) [17]. Deepwalk [38] and Node2vec [39] obtain the wandering sequence by wandering nodes. The sequence is obtained by the word2vec model to obtain the the embedding of nodes. In message passing, as in Equation (1), by multiplying the embedding of the first-order neighbor nodes and the weights of the edges, we obtain the the value of a message. The messages flow in the same direction as the transaction, from node S to node R. We obtain the embedding of the central node by aggregating all received message values. Combining the features of the neighboring nodes of a node is used to generate the features of that node, which is the method used for convolution in the spatial domain [40].

$$f_i = \max_{j \in N(i)} \left( f_j * W_{i,j} \right) \tag{1}$$

where $W_{i,j}$ is the weight between node $i$ and node $j$. $f_i$ is the embedding of node $i$, and $N(i)$ means the neighbor nodes of node $i$. In the end, every node aggregates its neighbor nodes as well as the network structure to obtain the embedding of the node.

(4) Detection: Based on the calculated node embedding values as the nodes' feature, we randomly select 80% labeled nodes and their first-order neighbors as training samples and the rest as test samples to test our phishing detection model. We only use the classification values of the labeled central nodes for comparison with the labeled values of the nodes.

First, we construct a large-scale Ethereum transaction network with labels by combining transaction records; subsequently, we randomly initialize the node embedding values and denote the embedding dimension of each node by d, to be used as the input to the model. After message passing, we obtain the embedding of the graph and nodes. After obtaining the embedding value of each node, we use GCN as the supervised learning method to train our model by fitting the embedding to the labeled data. After training, we validated the performance of the model with a test set based on the trained model.

### 3.3. Message Passing Based GCN

The GCN is a graph embedding method [41]. The graph convolution can be treated as message passing [42], whose main function is to obtain information from the surrounding neighboring nodes and neighboring edges, and finally, combine all the messages to obtain the embedding of the nodes. Message passing can be segregated into a message function, and a reduce function, and the process is shown in Figure 3.

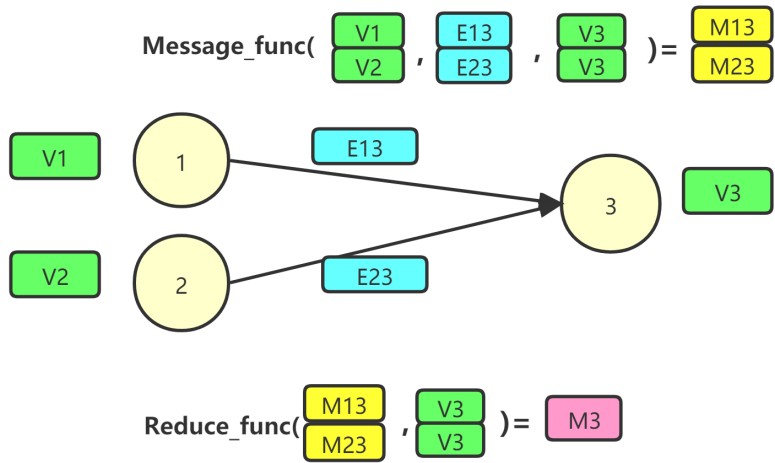

**Figure 3.** Message Passing of GCN.

**Message function**: The purpose of message passing is to pass the information required by the node for computation. The information of the node is the embedding value of the node, and the information of the edge is the transaction amount and transaction timestamp of the edge. For each edge, the source node of each edge passes its own information (V1 and V2) and the information of the edge (E13 and E23) to the destination node (V3), and for each destination node, it may receive multiple messages (M13 and M23) from the source nodes. Each destination node may receive messages (M13 and M23) from multiple source nodes, and these messages are then stored in a "mailbox".

**Reduce function**: The node updates its own node information based on the messages passed by its neighbors and performs a nonlinear transformation. Each node first aggregates the messages (M13 and M23) passed by the message function from the mailbox and empties the messages in the mailbox. Subsequently, it updates the node information (M3) by combining the aggregated results with the original information of the node.

For the message function, the source node data, the destination node and the edge data are aggregated into a message that is then stored inside the mailbox. The destination node of the edge, which is the sink node, updates its own node information based on the message received by the mailbox.

For nodes with multiple incoming degrees, each edge generates one message, and eventually, the sink node aggregates all the messages and updates them to its own node.

Similarly, for a node with multiple outgoing degrees, its information must be diffused outward by multiple edges. The mathematical formula for the messaging is as follows:

$$m_e^{(t+1)} = \varphi(x_v^t, x_u^t, w_e^t), (u, v, e) \in \epsilon \tag{2}$$

$$x_v^{(t+1)} = \psi(x_v^t, \rho(\{m_e^{(t+1)} : (u, v, e) \in \epsilon\})) \tag{3}$$

where $t$ is a state before sending messages. $x_i$ is the embedding of the i-th node, $w_e$ is the feature of the $e$-th edge. $\varphi$ is a message function defined on each edge that computes the source node $x_v$, $x_u$ which is the destination node, and the information of the edge $w_e$ as the message $m_e$. $\epsilon$ stands for the set of the nodes ($u$ and $v$) and edges ($e$). The aggregation function $\rho$ aggregates the messages received by the neighbor nodes. The update function $\psi$ updates the embedding of the node by combining the aggregated messages with the embedding of the node.

### 3.4. Structure of MP-GCN

In the input layer, the number of input neurons equals the node embedding dimension d [43], and the node embedding is randomly initialized before input. Next, we set a layer of graph convolution to compute the new embedding values for each node, using a fully connected layer for linear transformation. We set the hidden layer neurons equal to 8, use relu as the activation function, and set the dropout rate equals 0.5. After that, we use a layer of graph convolution to compute the node embedding values. We ended up with a linear layer to the output layer, and we set output neurons equal to 2 because the problem can be treated as a binary problem. The structure of our model is shown in Figure 4.

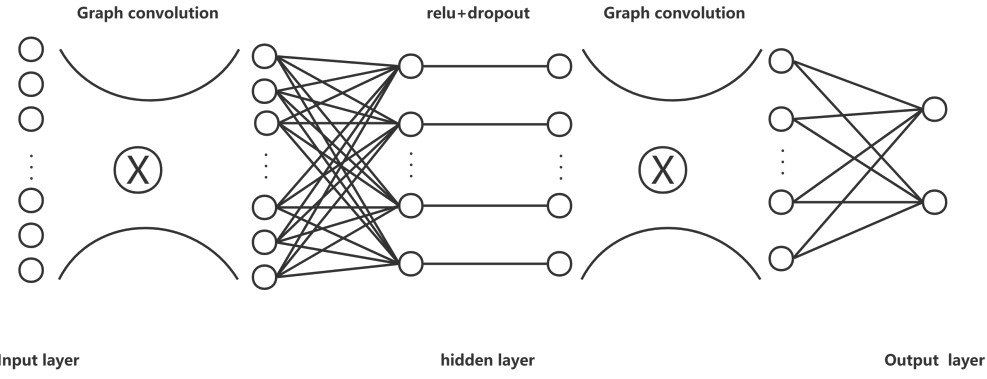

**Figure 4.** The structure of MP-GCN.

### 3.5. Detection Algorithm

The process for detecting phishing notes is described in Algorithm 1.

In our experiments, we make the value of weights ($W_e$) to be B or T between two nodes. First, we created a network based on transaction records and processed the data in a situation suitable for the MP-GCN (lines 1–3). Prior to training, the MP-GCN model must be initialized based on the defined network layers and the propagation function (lines 4–10). After instantiating the MP-GCN model, we trained the model using training data and tested it using test data (lines 11–18). The G[labeled] means the specific nodes' embedding and their labels and Y[labeled] means the specific nodes' label. We randomly added 1032 labeled phishing nodes and 968 labeled normal nodes (80% of all labeled nodes) and their first-order neighbor to the network as the training set, trained the internal parameters of the model based on the labeled values, and used the trained model to test against the nodes to distinguish phishing nodes from other nodes. We use the classification values of the labeled nodes to compare with the actual label values.

---

**Algorithm 1** Phishing nodes detection algorithm

---

**Input:** Transaction Records(*S, R, T, B*), labels of the sets *Y*, training epochs *epochs*, labeled
   train nodes' index [*labeled*], Normalized function *softmax*, loss function *nll_loss*
**Output:** The predicted values in the test set.
 1: Create transaction graph G from Records(*S, R, T, B*)
 2: Randomly initialize the embedding of each node
 3: Adding a self-loop to each node
 4: **Define** *graph_net*(*G*)
 5:    **for** $V_i$ in G **do**
 6:       $message \leftarrow \phi(V_i, neighbor(V_i), W_e)$
 7:       $V_i \leftarrow \psi(V_i, \rho(message))$
 8:    **end for**
 9: **end Define**
10: Instantiated models: *graph_net*()
11: **for** *epoch* = 0 to *epochs* **do**
12:    **for** *nodes, labels* in G[*labeled*] **do**
13:       $logp \leftarrow softmax(graph\_net(G), 1)$
14:       $loss \leftarrow nll\_loss(logp[labeled], Y[labeled])$
15:       *loss.backforward*
16:    **end for**
17:    predict()
18: **end for**

---

### 3.6. Implementation Details

We build the network based on the transaction records. The edge attributes amount and timestamp are equated to the attribute values in the transaction record, and the node's label attribute value is based on whether the node is a fishing node or not. In addition, the value of the node's label attribute also determines whether it will be used for evaluation.

After the network is transformed into a DGL class graph, the node embedding attribute values are obtained randomly, and the embedding attribute dimension is equal to the number of input neurons and the number of output neurons is equal to 2. We set up two layers of GCN, and the number of output neurons of the first GCN layer is equal to the number of hidden layer neurons, followed by a layer of relu activation function and a dropout with *p* = 0.5. Then followed by another layer of GCN with output dimension equal to 2. We use 80% of the labeled nodes and their first-order neighbor nodes as training nodes and pair the model output values with the label values. In the evaluation, we use the torch.max() function to obtain the classification values to compare with the label values. In the calculation of the confusion matrix, we use the proportion of true positives instead of the number of true positives. In the calculation of the confusion matrix, we use the proportion of true positives instead of the number of true positives. In calculating the roc curve, we use the softmax function to calculate the output result probability of the labeled nodes.

### 4. Experiments

In this section, we present the experimental results obtained using the proposed phishing detection model and discuss the parameters used in the model. By comparing the results with those obtained using other methods and different parameters, we evaluated our phishing detection model precision on Ethereum and the effects of the parameters. Our experimental environment is the following: the CPU is Core™ i5-9400 @ 3.6 GHz, the operating system is Windows 10, the RAM is 16GB, the Python version is 3.8, the deep learning library is DGL (v0.6.x) by using PyTorch as the backend.

### 4.1. Dataset

Our data are sourced from XBlock (http:/xblock.pro/#/search?types=datasets) (accessed on 1 January 2022), which includes the labeled addresses and the first-order transaction records of the labeled addresses. Each record means a transaction between two different addresses. In the dataset, 1262 addresses are labeled as phishing addresses and 1222 addresses are normal addresses. The data ratio of 1:1 is to reduce the impact of positive and negative imbalance. We build a transaction network from the records. In this network, each node represents an address, each edge represents a pair of nodes between Ether transactions and contains the total transmission amounts. Thus our transaction network includes 100,000 nodes and 250,000 edges. Each node contains embedding information of dimension d. The value of the label attribute is set to 1 for phishing nodes and $-1$ for normal nodes. The first-order neighborhood node attributes of the phishing and normal nodes are set to 0. The purpose of the label attribute value is to filter out the nodes that need to be analyzed and their actual attributes for evaluation. Finally, we input the entire network information and used 80% of the labeled data as training data to train the model and the remaining as test data. We only use the labeled nodes for analysis.

### 4.2. Evaluation Metrics

To evaluate the performance of different methods in phishing detection, we considered three evaluation metrics: precision, recall, and F-score. Because the values of initialized nodes embedding come from standard normal distribution N(0, 1), we repeated 10 times using different values of nodes embedding to reduce the errors and obtained the average results.

The three metrics are defined as follows:

$$Precision = \frac{TP}{TP + FP} \tag{4}$$

$$Recall = \frac{TP}{TP + FN} \tag{5}$$

$$F\text{-}score = 2 \times \frac{Precision \times Recall}{Precision + Recall} \tag{6}$$

where $TP$ is True Positive which means positive samples are correctly classified, $TN$ is True Negative which means negative samples are correctly classified. $FP$ is False Positive, which means positive samples are incorrectly classified, and $FN$ is False Negative, which means negative samples are incorrectly classified.

Precision refers to the number of positive samples correctly classified as a percentage of all positive samples. Recall is the proportion of samples correctly classified as positive for all samples classified as positive. The F-score score is the harmonic mean of the precision and recall. All evaluation metrics are in the range of [0, 1]. The closer the metric to 1, the better is the classification performance.

### 4.3. Performance and Comparisons

In our experiments, we randomly segregated the dataset into a training set and a test set at a certain ratio (rate = 80%). Each labeled node and its first-order neighbor nodes are used as samples. For the classification model, the state of the labelled node should be distinguished; therefore, we set the output dimension to two. The input dimension $d$ = {4, 8, 16, 32, 64} and the nodes embedding in dimension d are randomly initialized by using module nn.Embedding from Pytorch, which is for saving word embedding [44]. In the forward propagation function, we used one of four different aggregation functions {sum, max, min, mean} to aggregate the nodes' embedding, and we used one of two features {timestamp and amount} as the weights of the edges. We used relu, which is an activation function to fit the nonlinear relationship, and set the dropout rate $p$ = 0.5. The $p$ is a parameter to prevent overfitting and improve model generalization. For the model training,

we used Adam [45] as the loss optimizer and the binary cross-entropy loss as the loss function with a learning rate of 0.01.

In our experiments, to demonstrate the prediction precision of our MP-GCN, we compare our method with the following methods:

**Deepwalk** [38]: The sequence of the node's neighboring nodes is obtained using the random walk strategy, which consequently yields structural information.

**Node2vec** [39]: Based on Deepwalk, wandering is specified such that a tradeoff can be achieved between depth-first and breadth-first.

**Trans2vec** [25]: Based on Node2vec, the best attribute value is obtained by balancing the weights of two attributes (amount and timestamp) based on the two attributes of the edge.

**Timestamp-GCN**: One of our methods, represents a GCN with the timestamp as the weight.

**MP-GCN**: Our method represents a GCN with the amount as the weight.

Our experiments set embedding dimension d = 32, epoch = 500, and the sum as the aggregation function to obtain the best experimental results. The comparison with other embedding methods is shown in Table 1. Deepwalk and Node2vec consider only the structural information of the graph thus their classification precision rate is less an 87.0%. The method Trans2vec, which considers the transaction amount and timestamp, yielded classification precision approximately 92.7%. Deepwalk and Trans2vec only use the characteristics of the neighboring nodes of the nodes considered; they do not consider the overall network topology information and cannot fully utilize the node characteristics in the transaction network. The classification precision of the MP-GCN, which considers the transaction amount and structural information, can reach 93.5%, which is the best performance achieved hitherto. The Timestamp-GCN, which uses timestamp as the transaction weight, its classification performance is not as good as above, is only 73.2%.

**Table 1.** Classification performance of different methods.

| Method | Precision | Recall | F-Score |
|---|---|---|---|
| Deepwalk | 0.799 | 0.762 | 0.780 |
| Node2vec | 0.870 | 0.822 | 0.845 |
| Time-base Bias | 0.864 | 0.822 | 0.842 |
| Amount-base Bias | 0.883 | 0.855 | 0.868 |
| Trans2vec | 0.927 | 0.893 | 0.908 |
| Timestamp-GCN | 0.732 | 0.703 | 0.716 |
| **MP-GCN** | **0.935** | **0.904** | **0.919** |

*4.4. Impact of Embedding Dimension*

The embedding dimension refers to how many factors are used to represent the features of a node. We set sum as the aggregation function and epoch = 500 in this experiment.

To study the impact of the embedding dimension, we initialized the node embedding in d = 4, 8, 10, 16, 32, 64 in Figures 5–7; the larger the node vector dimension, the better is the classification performance. For d > 16, the precision rate by using our model increased slowly or even remains stable. Because the increase in dimensionality can retain a richer network structure and more node information, the generalization ability of the model can be enhanced. However, after the dimensionality increases to a certain level, the model will be overfitted. The accuracy of Trans2vec increases with dimensionality but still cannot break 93%. Hence, a d in our model between 16 and 32 yielded favorable results in terms of the model classification metrics.

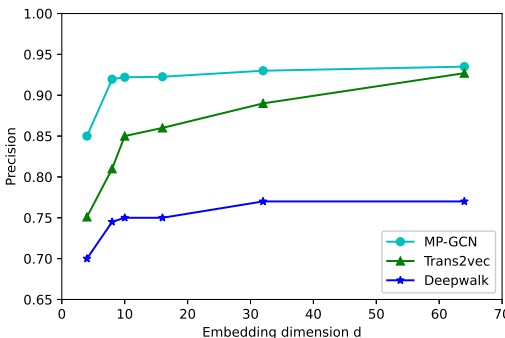

**Figure 5.** Precision of the classification.

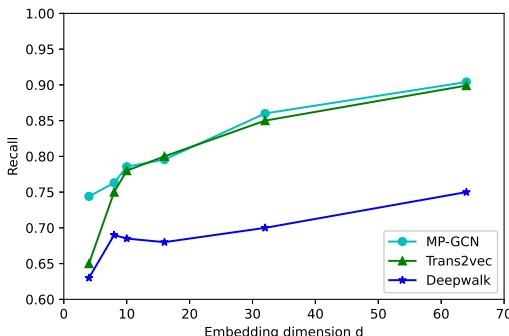

**Figure 6.** Recall of the classification.

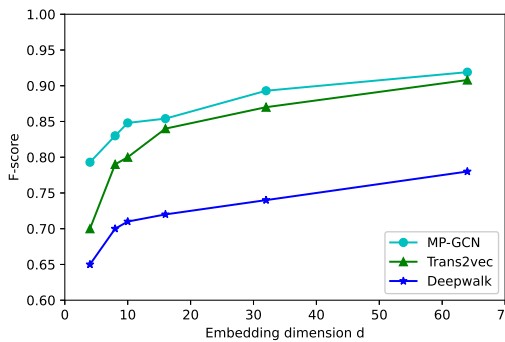

**Figure 7.** Fscore of the classification.

*4.5. Impact of Aggregation Functions*

The aggregation function is used to aggregate embedding from their neighboring nodes and the features from their connected edges as the center node's embedding. The performance is shown in Figure 8.

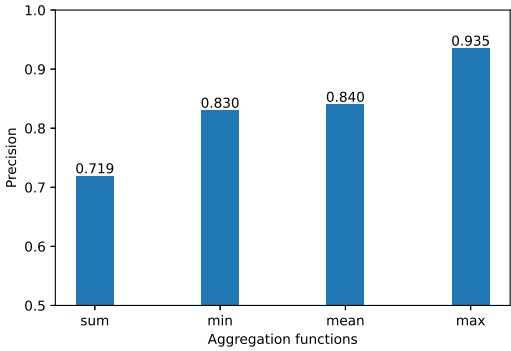

**Figure 8.** Impact of different aggregation functions.

We set d = 16 and epoch = 500, the sum aggregation function was used to sum all accepted data. A node with a high degree is likely to generate a large aggregation vector, whereas a node with a low degree tends to yield a small aggregation vector, which is particularly susceptible to the degree size. In addition, the mean was used to obtain the mean data from the "mailbox". Thus sum and mean are not appropriate. The max aggregation function, which is used to extract the maximum value of the accepted data, is relatively large and is particularly sensitive to the transaction amount. That phenomenon is consistent with the characteristics of money transfer in phishing fraud. The min aggregation function, which is used to extract the minimum value of the accepted data, is relatively small to the transaction amount.

### 4.6. Impact of Training Epoch

The training epoch refers to the number of parameter adjustments within the model. The parameters are adjusted by performing loss calculations with samples and labels for each training.

Figure 9 shows the distribution of the precision rate against the gradually increasing number of training rounds. During training, the training precision can be improved rapidly based on a small number of training sessions. However, after 800 training sessions, it is overfitted gradually. For a relatively small graph, excessive training will adversely affect the classification precision.

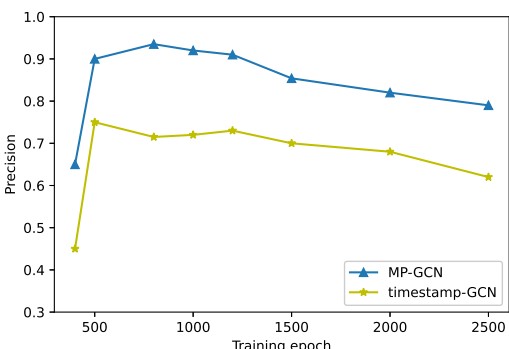

**Figure 9.** Impact of training epoch.

During training, the increase of training rounds will improve the model's ability to fit the dataset and continuously adjust the parameters of the model so that the structural and transactional characteristics of the data are fully considered, which satisfies our detection requirements. Within 100 to 500 rounds of training, the model affords a higher and more stable classification precision; continuing training on this basis will result in parameter overfitting and a gradual decrease in precision.

### 4.7. Stability of the Experiment

By analyzing the confusion matrix in Figure 10, we found that 96% of the positive cases were correctly classified from the positive cases and 89% of the negative cases were correctly classified. Our model can accurately filter out the real phishing nodes from the suspected phishing nodes. But there is an 11% probability of misclassifying normal nodes as phishing nodes because their network structure may be very similar to that of phishing nodes. The ROC curve (Receiver operating characteristic curve) graph is shown in Figure 11, it shows the classification performance of our model, our AUC (Area under the ROC curve) value is 0.92, which is a very good classification model.

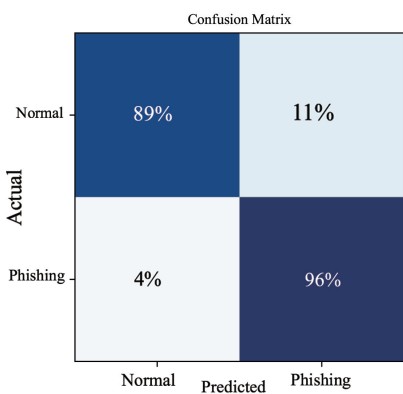

**Figure 10.** Confusion matrix for classification results.

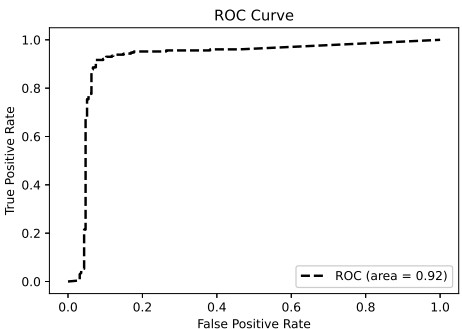

**Figure 11.** Classification of roc curves.

*4.8. Sample Analysis of Misclassified*

Figure 12 shows a sample of normal nodes that were misclassified as phishing nodes. Some of these normal nodes have no or few reverse interactions with incoming nodes but transact with a large number of nodes, and some have few total transactional edges to form an accurate classification of nodes. In short insufficient number of transacting edges may not result in an accurate classification, and insufficient interaction between nodes may also allow the model to produce incorrect judgments. Therefore, in practice, even if nodes are classified as phishing nodes, users may still need to make further judgments based on the number of transactions and interactions.

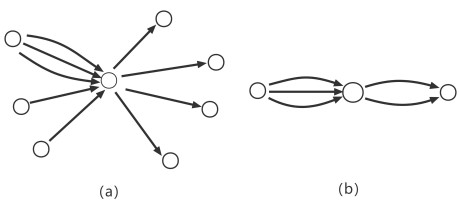

**Figure 12.** Misclassification samples: (**a**) is an example with few reverse interactions between nodes, and (**b**) is an example with a relatively small number of nodes.

**5. Discussion and Future Works**

Our experiments performed well, but still presented a few disadvantages. For example, we could not fully utilize the timestamp and amount weights. Thus, in the future, we would use the multi-weighting method to fully consider timestamp and amount to improve the classification precision. Furthermore, We did not know how to initialize the node embedding to obtain the best performance; thus, we had to randomly experiment 10 times to reduce its effect. In addition, phishing scams were detected based on existing partial transaction records. In other words, the graphs used for training were static and could not be used to analyze dynamic graph information with a time sequence for the dynamic

prediction of nodes. The next step is to consider the prediction of phishing nodes in the order of transactions added in the time sequence.

## 6. Conclusions

Herein, we proposed an MP-GCN model for phishing fraud detection in Ethereum networks. We first constructed the phishing network and divided it into a training set and a test set. Then we use the MAX function as the aggregation function to improve the model fitting ability. Meanwhile, the embedding dimension of nodes is directly related to the classification performance, but too high dimensionality tends to lead to overfitting. Finally, the number of training is in the appropriate range to obtain the desired results. We confirmed the effectiveness of our proposed method by comparing the results obtained with those of the corresponding recorded test set. Our model outperforms conventional general methods for phishing node detection in Ethereum transaction networks.

**Author Contributions:** Methodology, software, and validation: T.Y., J.X., X.C. and Z.X.; investigation, data curation and supervision: J.X.; review, editing, and supervision: T.Y., J.X. and Z.X.; funding acquisition: J.X. and X.C. All authors read and agreed to the published version of the manuscript.

**Funding:** This research was financially supported by the National Natural Science Foundation of China (No.61702318), Guangdong province special fund for science and technology ("major special projects + task list") project (No. STKJ2021201), 2020 Li Ka Shing Foundation Cross-Disciplinary Research Grant (No. 2020LKSFG08D), Special Projects in Key Fields of Guangdong Universities (No. 2020ZDZX3073), Research on Food Production and Marketing traceability Software system based on Blockchain (No. STKJ2021011), the Shantou University Scientific Research Foundation for Talents (NTF18022), and in part by Guangdong Basic and Applied Basic Research Foundation (No. 2021A1515012527).

**Institutional Review Board Statement:** Not applicable.

**Informed Consent Statement:** Not applicable.

**Data Availability Statement:** Not applicable.

**Acknowledgments:** The authors thank Sun Yat-sen University, inplus Labs for providing the dataset, and gratefully acknowledge all anonymous reviewers and editors for their constructive suggestions for the improvement of this paper.

**Conflicts of Interest:** The authors declare no conflict of interest.

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
