# Peer review of "MP-GCN: A Phishing Nodes Detection Approach via Graph Convolution Network for Ethereum"

_applsci, doi:10.3390/app12147294_

Round 1

Reviewer 1 Report

This paper proposes a phishing node detection approach for ETH. It is required by blockchain communities. comments:

  1. the method needs to be given more details such as implementation details.
  2.  You need to upload your source code into Github or opensource version control system in order to publish your paper. 

Author Response

We are very grateful to Reviewer for reviewing the paper so carefully. We have carefully considered the suggestion of Reviewer and make some changes.

Point 1:

the method needs to be given more details such as implementation details.

Response1:

You make a very sensible point, and we do need to add some implementation details. We have added a subsection 3.6 Implementation details in line 297-315(page8). In that subsection we describe how to create a transaction network from transaction records, how to set the attributes of edges, how to set the attributes of nodes, and what the node attribute values represent. We also describe how the initial embedding values of the nodes are set and how the embedding dimension d is set. In the model, the number of input and output neurons is explained, the details of the sequence of how the model is built and the setting of the dropout value p. We explain how to obtain the output values for comparison with the labels.

Point2:

You need to upload your source code into Github or opensource version control system in order to publish your paper. 

Response2:

We all agree that open our code would be useful to study. Now, we have upload our code on github(https://github.com/zzykt/MPGCN_A_Phishing_Nodes_Detection_Approach), we have performed a series of explanations above, including details of the data processing for description, and we have uploaded the trained model for easy testing.

Reviewer 2 Report

The paper investigates an interesting problem – how to detect nodes in Ethereum network, which potentially are gaining from Phishing attacks or other malicious activities. Authors believe the pattern of incoming and outgoing transactions are different between nodes, which are gaining from Phishing attacks and legally acting user nodes. This assumption does not take into account the same pattern can be noticed if the user gains from different malicious activities, for example ransomware, or event provides some kind of services, where payment is done but the transfers out are rarer. Now the dataset is taken, where Phishing and legitimate user nodes are presented, but no analysis on the labeling properties are presented.

The title “A Phishing Nodes Detection Approach …” is misleading as the nodes do not execute Phishing attack, it is just a source where the Phishing attack results (Ethers, gained from Phishing affected users) are transferred. The proposed solution will not prevent Phishing attacks, only the gaining e-wallets can be identified. As well the nodes will be identified only after some time, when enough of victims will do the transactions. The need of incoming links is not analyzed in the search; therefore, it is not clear how time-vise affective the solution can be (maybe the victims could report the accident sooner than the method will be able to identify these nodes).

Regarding the paper quality, it lacks a deeper analysis or related works. For example, no Blockchain node, executing malicious activities or frauds, detection methods were analyzed in the related work section. Meanwhile some similar papers exist and the authors reference to one of them in Section 4.3. Therefore, it is difficult to understand what is the scientific novelty of the paper. Basically the paper proposes to use a different weight function. So the analysis of related works and highlight of scientific novelty must be highlighted.

The used dataset and research results are not presented with enough details as well. Authors state “In the dataset, 1660 addresses are labeled as phishing addresses and 1700 addresses are normal addresses”. The dataset is taken from XBlock, while they are stating “As 1259 addresses are labeled as phishing nodes which are the targets of the detection approach, we randomly select 1259 unlabeled nodes as the outliers”. Therefore, it is unclear how the additional addresses were obtained. Additionally, it is not clear is the even distribution between classes are guaranteed, as after building a graph of the dataset data, “transaction network includes 100,000 nodes and 220,000 edges”. If all nodes are analyzed, the phishing nodes would present less than 20% of the nodes.

The results are presented as precision, recall and F-score metrics. No analysis on failed classification cases are presented. The lack of confusion matrix, Area Under the Curve (AUC), standard deviation between achieved results do not allow the reader to understand the stability of the solution and suitability to be applied.

It is nice some additional experiments were executed to find out the impact of embedding dimension, aggregation function and training epochs. However, the results are limited, as only precision is compared, no analysis on training and application time, misclassified cases investigation are executed.

In my opinion the paper in its current state is not suitable for publishing in this journal and need a major revision.

Author Response

We are very grateful to Reviewer for reviewing the paper so carefully. We have carefully considered the suggestion of Reviewer and make some changes.

Point1:

The paper investigates an interesting problem – how to detect nodes in Ethereum network, which potentially are gaining from Phishing attacks or other malicious activities. Authors believe the pattern of incoming and outgoing transactions are different between nodes, which are gaining from Phishing attacks and legally acting user nodes. This assumption does not take into account the same pattern can be noticed if the user gains from different malicious activities, for example ransomware, or event provides some kind of services, where payment is done but the transfers out are rarer. Now the dataset is taken, where Phishing and legitimate user nodes are presented, but no analysis on the labeling properties are presented.

Response1:

Thank you for your insightful comments. Our experiment is to explore how to classify phishing nodes and non-phishing nodes from the difference of transaction behavior patterns. If non-phishing nodes are involved in other classes of anomalies (e.g., ransomware), which may affect our classifier results, later, if the conditions are ripe, we may conduct the relevant research. The data provided by xblock can help us classify phishing and non-phishing nodes.

We added the description of node labels in subsection 4.1 in line 334 - 340 (page9), that is, the value of the label attribute of each phishing node is set to 1, and the value of the label attribute of each normal node is set to -1. The first-order neighborhood node attributes of phishing and normal nodes are set to 0. We emphasized in the modified version that the purpose of the label attribute values is to filter out the nodes that need to be analyzed and their actual for evaluation.

Point2:

The title “A Phishing Nodes Detection Approach …” is misleading as the nodes do not execute Phishing attack, it is just a source where the Phishing attack results (Ethers, gained from Phishing affected users) are transferred. The proposed solution will not prevent Phishing attacks, only the gaining e-wallets can be identified. As well the nodes will be identified only after some time, when enough of victims will do the transactions. The need of incoming links is not analyzed in the search; therefore, it is not clear how time-vise affective the solution can be (maybe the victims could report the accident sooner than the method will be able to identify these nodes).

Response2:

As Reviewer suggested that It is important to explain clearly the Detection word. In practice, a user node wants to transact with a node, either the user queries whether it is a phishing node from a list published by a third-party authority or the user is completely unknown. In the case of not knowing whether the node is a phishing node, the output is used to determine whether the node is a phishing node or not by inputting the transaction network consisting of the destination node and its first-order or second-order neighbors. Therefore, we use “A Phishing Nodes Detection Approach” as the title. Our experiments improve the timeliness of detecting phishing nodes because users do not need to wait for the authority to release the latest list of phishing nodes. We explain the content in line 61 - 68 (page 2). In our validation experiments, we assume that we do not know whether the central node is a phishing node or not, and after the algorithm obtains a prediction or classification value, then we compare the value with the real value of the node to verify the accuracy of our model. Also we have added the purpose statement in line 114 (page 3) in the revised version to illustrate these contents.

Point3:

Regarding the paper quality, it lacks a deeper analysis or related works. For example, no Blockchain node, executing malicious activities or frauds, detection methods were analyzed in the related work section. Meanwhile some similar papers exist and the authors reference to one of them in Section 4.3. Therefore, it is difficult to understand what is the scientific novelty of the paper. Basically the paper proposes to use a different weight function. So the analysis of related works and highlight of scientific novelty must be highlighted.

Response3:

Thank you for your comments. We agree that it is necessary to add some detection methods for malicious activities or fraud in Ethereum from line 150 to 168 (page 4). The introduction section of our original draft of this paper did not make the novelty and importance clear. In view of this, we have strengthened the introductory section to highlight the innovation in line 107-116 (page 3). The innovation of this paper lies in the fact that, we propose a new graph convolution method that not only takes into account the overall topological information of the network, but also makes full use of the node features in the transaction network, the method outperforms existing methods in the ethereum phishing detection problem, and we provide a way to perform phishing detection on unknown nodes, and users can determine faster whether the node is abnormal or not without waiting for a third-party organization to publish a list of phishing nodes. What’s more, we discuss the effect of model structure on the classification results by discussing different model parameters. The stability of our model is also analyzed.

Point4:

The used dataset and research results are not presented with enough details as well. Authors state “In the dataset, 1660 addresses are labeled as phishing addresses and 1700 addresses are normal addresses”. The dataset is taken from XBlock, while they are stating “As 1259 addresses are labeled as phishing nodes which are the targets of the detection approach, we randomly select 1259 unlabeled nodes as the outliers”. Therefore, it is unclear how the additional addresses were obtained. Additionally, it is not clear is the even distribution between classes are guaranteed, as after building a graph of the dataset data, “transaction network includes 100,000 nodes and 220,000 edges”. If all nodes are analyzed, the phishing nodes would present less than 20% of the nodes.

Response4:

We apologize for the error in the data, we mistook all 1660 nodes from the k2 order as phishing nodes, we only included 1262 nodes in the actual experiment, as the remaining 400 nodes are not in the list of label nodes provided by the dataset, so they will not be added to the experimental for validation. In the latest manuscript we corrected the data to 1262 phishing nodes and 1222 normal nodes. 1262 nodes are directly provided by this dataset (although xblock claims there are 1259 nodes) and 1222 normal nodes are randomly selected from the normal 1700 nodes of the K2 dataset. At the end of subsection 4.1 in line 339, we added: We use only the labeled nodes for analysis. In our experiments there are 2484 labelled nodes, of which 1262 nodes are phishing nodes and 1222 nodes are normal nodes. The training set has 2000 nodes, of which 1032 nodes are phishing nodes and 968 nodes are normal nodes. Therefore the phishing node only accounts for 50%.

Point5:

The results are presented as precision, recall and F-score metrics. No analysis on failed classification cases are presented. The lack of confusion matrix, Area Under the Curve (AUC), standard deviation between achieved results do not allow the reader to understand the stability of the solution and suitability to be applied.

Response5:

We agree that more study or more data would be useful to research. In the revised version, we added confusion matrix and ROC curve and AUC area in subsection 4.7 in line 439-446, and we analyzed some reasons for misclassification in subsection 4.8 in line 447-455. The confusion matrix show that our model can have 96% accuracy accurately filter out the real phishing nodes from the suspected phishing nodes. But there is an 11% probability of misclassifying normal nodes as phishing nodes because their network structure may be very similar to that of phishing nodes.

Point6:

It is nice some additional experiments were executed to find out the impact of embedding dimension, aggregation function and training epochs. However, the results are limited, as only precision is compared, no analysis on training and application time, misclassified cases investigation are executed.

Response6:

Thank you very much for your very profound suggestions. In the revised version, we have added a survey and analysis of misclassifications performed in subsection 4.8. Some of these misclassified normal nodes have no or few reverse interactions with incoming nodes but transact with a large number of nodes, and some have few total transactional edges to form an accurate classification of nodes. Therefore, in practice, insufficient transaction edges or insufficient interactions between nodes can affect the classification results of the model. The users may still need to make further judgments based on the number of transactions and interactions.

Reviewer 3 Report

1. The research is significant, however, needs to do minor adjustments in Fig 5 which is too small and not clear.

Author Response

We are very grateful to Reviewer for reviewing the paper so carefully. We have carefully considered the suggestion of Reviewer and make some changes.

Point1:

The research is significant, however, needs to do minor adjustments in Fig 5 which is too small and not clear.

Response1:

Thank you very much for your correction, and we agree that too small figures are not suitable for readers. In the revised version, we have modified the figure size and layout so that the figure can be seen clearly.

Best regard!

Round 2

Reviewer 2 Report

I still see some places to improve, for example more detailed description of the dataset (statistics of incoming and outgoing neighbors for each class, etc.), more detailed analysis of the results, explanations of the images, etc.

However there were improvements after the last review, which are enough to accept the paper for publishing